# Online Continual Learning with Maximally Interfered Retrieval

**Rahaf Aljundi**[*]
KU Leuven
rahaf.aljundi@gmail.com

**Lucas Caccia**[*]
Mila
lucas.page-caccia@mail.mcgill.ca

**Eugene Belilovsky**[*]
Mila
eugene.belilovsky@umontreal.ca

**Massimo Caccia**[*]
Mila
massimo.p.caccia@gmail.com

**Min Lin**
Mila
mavenlin@gmail.com

**Laurent Charlin**
Mila
lcharlin@gmail.com

**Tinne Tuytelaars**
KU Leuven
tinne.tuytelaars@esat.kuleuven.be

## Abstract

Continual learning, the setting where a learning agent is faced with a never ending stream of data, continues to be a great challenge for modern machine learning systems. In particular the online or "single-pass through the data" setting has gained attention recently as a natural setting that is difficult to tackle. Methods based on replay, either generative or from a stored memory, have been shown to be effective approaches for continual learning, matching or exceeding the state of the art in a number of standard benchmarks. These approaches typically rely on randomly selecting samples from the replay memory or from a generative model, which is suboptimal. In this work we consider a controlled sampling of memories for replay. We retrieve the samples which are most interfered, i.e. whose prediction will be most negatively impacted by the foreseen parameters update. We show a formulation for this sampling criterion in both the generative replay and the experience replay setting, producing consistent gains in performance and greatly reduced forgetting. We release an implementation of our method at https://github.com/optimass/Maximally_Interfered_Retrieval.

## 1 Introduction

Artificial neural networks have exceeded human-level performance in accomplishing individual narrow tasks [19]. However, such success remains limited compared to human intelligence that can continually learn and perform an unlimited number of tasks. Humans' ability of learning and accumulating knowledge over their lifetime has been challenging for modern machine learning algorithms and particularly neural networks. In that perspective, continual learning aims for a higher level of machine intelligence by providing the artificial agents with the ability to learn online from a non-stationary and never-ending stream of data. A key component for such never-ending learning

---

[*]Authors contributed equally

process is to overcome the catastrophic forgetting of previously seen data, a problem that neural networks are well known to suffer from [13]. The solutions developed so far often relax the problem of continual learning to the easier task-incremental setting, where the data stream can be divided into tasks with clear boundaries and each task is learned offline. One task here can be recognizing hand written digits while another different types of vehicles (see [24] for example).

Existing approaches can be categorized into three major families based on how the information regarding previous task data is stored and used to mitigate forgetting and potentially support the learning of new tasks. These include *replay-based* [8, 30] methods which store prior samples, *dynamic architectures* [35, 39] which add and remove components and *prior-focused* [18, 41, 9, 7] methods that rely on regularization.

In this work, we consider an online continual setting where a stream of samples is seen only once and is not-iid. This is a much harder and more realistic setting than the milder incremental task assumption[4] and can be encountered in practice e.g. social media applications. We focus on the *replay-based* approach [26, 36] which has been shown to be successful in the online continual learning setting compared to other approaches [26]. In this family of methods, previous knowledge is stored either directly in a replay buffer, or compressed in a generative model. When learning from new data, old examples are reproduced from a replay buffer or a generative model.

In this work, assuming a replay buffer or a generative model, we direct our attention towards answering the question of *what samples should be replayed from the previous history when new samples are received*. We opt for retrieving samples that suffer from an increase in loss given the estimated parameters update of the model. This approach also takes some motivation from neuroscience where replay of previous memories is hypothesized to be present in the mammalian brain [27, 34], but likely not random. For example it is hypothesized in [15, 25] similar mechanisms might occur to accommodate recent events while preserving old memories.

We denote our approach Maximally Interfered Retrieval (MIR) and propose variants using stored memories and generative models. The rest of the text is divided as follows: we discuss closely related work in Sec. 2. We then present our approach based on a replay buffer or a generative model in Sec. 3 and show the effectiveness of our approach compared to random sampling and strong baselines in Sec. 4.

## 2 Related work

The major challenge of continual learning is the catastrophic forgetting of previous knowledge once new knowledge is acquired [12, 32] which is closely related to the stability/plasticity dilemma [14] that is present in both biological and artificial neural networks. While these problems have been studied in early research works [10, 11, 20, 21, 37], they are receiving increased attention since the revival of neural networks.

Several families of methods have been developed to prevent or mitigate the catastrophic forgetting phenomenon. Under the fixed architecture setting, one can identify two main streams of works: i) methods that rely on replaying samples or virtual (generated) samples from the previous history while learning new ones and ii) methods that encode the knowledge of the previous tasks in a prior that is used to regularize the training of the new task [17, 40, 1, 28]. While the prior-focused family might be effective in the task incremental setting with a small number of disjoint tasks, this family often shows poor performance when tasks are similar and training models are faced with long sequences as shown in Farquhar and Gal [9].

Replayed samples from previous history can be either used to constrain the parameters update based on the new sample, to stay in the feasible region of the previous ones [26, 6, 5] or for rehearsal [30, 33]. Here, we consider a rehearsal approach on samples played from previous history as it is a cheaper and effective alternative to the constraint optimization approach [8, 5]. Rehearsal methods usually play random samples from a buffer, or pseudo samples from a generative model trained on the previous data Shin et al. [36]. These works showed promising results in the offline incremental tasks setting and recently been extended to the online setting [8, 5], where a sequence of tasks forming a non i.i.d. stream of training data is considered with one or few samples at a time. However, in the online setting and given a limited computational budget, one can't replay all buffer samples each time and it

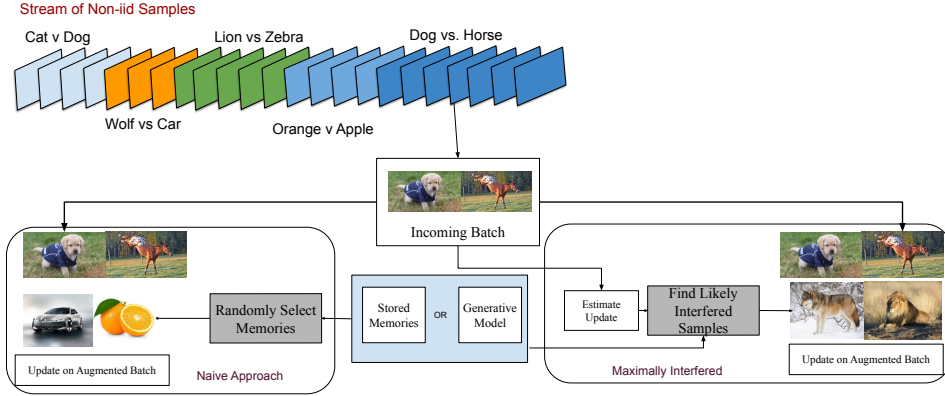

Figure 1: High-level illustration of a standard rehearsal method (left) such as generative replay or experience replay which selects samples randomly. This is contrasted with selecting samples based on interferences with the estimated update (right).

becomes crucial to select the best candidates to be replayed. Here, we propose a better strategy than random sampling in improving the learning behaviour and reducing the interference.

Continual learning has also been studied recently for the case of learning generative models [29, 22]. Riemer et al. [31] used an autoencoder to store compressed representation instead of raw samples. In this work we will leverage this line of research and will consider for the first time generative modeling in the online continual learning setting.

# 3 Methods

We consider a (potentially infinite) stream of data where at each time step, $t$, the system receives a new set of samples $\boldsymbol{X}_t, \boldsymbol{Y}_t$ drawn non i.i.d from a current distribution $D_t$ that could itself experience sudden changes corresponding to task switching from $D_t$ to $D_{t+1}$.

We aim to learn a classifier $f$ parameterized by $\theta$ that minimizes a predefined loss $\mathcal{L}$ on new sample(s) from the data stream without interfering, or increasing the loss, on previously observed samples. One way to encourage this is by performing updates on old samples from a stored history, or from a generative model trained on the previous data. The principle idea of our proposal is that instead of using randomly selected or generated samples from the previous history [6, 36], we find samples that would be (maximally) interfered by the new incoming sample(s), had they been learned in isolation (Figure 1). This is motivated by the observation that the loss of some previous samples may be unaffected or even improved, thus retraining on them is wasteful. We formulate this first in the context of a small storage of past samples and subsequently using a latent variable generative model.

## 3.1 Maximally Interfered Sampling from a Replay Memory

We first instantiate our method in the context of experience replay (ER), a recent and successful rehearsal method [8], which stores a small subset of previous samples and uses them to augment the incoming data. In this approach the learner is allocated a memory $\mathcal{M}$ of finite size, which is updated by the use of reservoir sampling [3, 8] as the stream of samples arrives. Typically samples are drawn randomly from memory and concatenated with the incoming batch.

Given a standard objective $\min_\theta \mathcal{L}(f_\theta(\boldsymbol{X_t}), \boldsymbol{Y}_t)$, when receiving sample(s) $\boldsymbol{X}_t$ we estimate the would-be parameters update from the incoming batch as $\theta^v = \theta - \alpha \nabla \mathcal{L}(f_\theta(\boldsymbol{X}_t), \boldsymbol{Y}_t)$, with learning rate $\alpha$. We can now search for the top-$k$ values $x \in \mathcal{M}$ using the criterion $s_{MI\text{-}1}(x) = l(f_{\theta^v}(x), y) - l(f_\theta(x), y)$, where $l$ is the sample loss. We may also augment the memory to additionally store the best $l(f_\theta(x), y)$ observed so far for that sample, denoted $l(f_{\theta^*}(x), y)$. Thus instead we can evaluate $s_{MI\text{-}2}(x) = l(f_{\theta^v}(x), y) - \min\big(l(f_\theta(x), y), l(f_{\theta^*}(x), y)\big)$. We will consider both versions of this criterion in the sequel.

We denote the budget of samples to retrieve, $\mathcal{B}$. To encourage diversity we apply a simple strategy of performing an initial random sampling of the memory, selecting $C$ samples where $C > \mathcal{B}$ before applying the search criterion. This also reduces the compute cost of the search. The ER algorithm with MIR is shown in Algorithm 1. We note that for the case of $s_{MI\text{-}2}$ the loss of the $C$ selected samples at line 7 is tracked and stored as well.

## 3.2 Maximally Interfered Sampling from a Generative Model

We now consider the case of replay from a generative model. Assume a function $f$ parameterized by $\theta$ (e.g. a classifier) and an encoder $q_\phi$ and decoder $g_\gamma$ model parameterized by $\phi$ and $\gamma$, respectively. We can compute the would-be parameter update $\theta^v$ as in the previous section. We want to find in the given feature space data points that maximize the difference between their loss before and after the estimated parameters update:

$$\max_{\boldsymbol{Z}} \mathcal{L}\big(f_{\theta^v}(g_\gamma(\boldsymbol{Z})), \boldsymbol{Y}^*\big) - \mathcal{L}\big(f_{\theta'}(g_\gamma(\boldsymbol{Z})), \boldsymbol{Y}^*\big) \quad \text{s.t.} \quad ||z_i - z_j||_2^2 > \epsilon \, \forall z_i, z_j \in \boldsymbol{Z} \text{ with } z_i \neq z_j \tag{1}$$

with $Z \in \mathbb{R}^{\mathcal{B} \times \mathcal{K}}$, $\mathcal{K}$ the feature space dimension, and $\epsilon$ a threshold to encourage the diversity of the retrieved points. Here $\theta'$ can correspond to the current model parameters or a historical model as in Shin et al. [36]. Furthermore, $y^*$ denotes the *true* label i.e. the one given to the generated sample by the real data distribution. We will explain how to approximate this value shortly. We convert the constraint into a regularizer and optimize the Equation 1 with stochastic gradient descent denoting the strength of the diversity term as $\lambda$. From these points we reconstruct the full corresponding input samples $\boldsymbol{X}' = g_\gamma(Z)$ and use them to estimate the new parameters update $\min_\theta \mathcal{L}(f_\theta(\boldsymbol{X}_t \cup \boldsymbol{X}'))$.

Using the encoder encourages a better representation of the input samples where similar samples lie close. Our intuition is that the most interfered samples share features with new one(s) but have different labels. For example, in handwritten digit recognition, the digit 9 might be written similarly to some examples from digits {4,7}, hence learning 9 alone may result in confusing similar 4(s) and 7(s) with 9 (Fig. 2). The retrieval is initialized with $\boldsymbol{Z} \sim q_\phi(\boldsymbol{X}_t)$ and limited to a few gradient updates, limiting its footprint.

To estimate the loss in Eq. 1 we also need an estimate of $y^*$ i.e. the label when using a generator. A straightforward approach for is based on the generative replay ideas [36] of storing the predictions of a prior model. We thus suggest to use the predicted labels given by $f_{\theta'}$ as pseudo labels to estimate $y^*$. Denoting $y_{pre} = f_{\theta'}(g_\gamma(z))$ and $\hat{y} = f_{\theta^v}(g_\gamma(z))$ we compute the KL divergence, $D_{KL}(y_{pre} \parallel \hat{y})$, as a proxy for the interference.

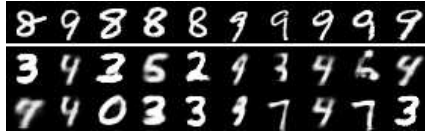

Figure 2: Most interfered retrieval from VAE on MNIST. Top row shows incoming data from a final task (8 v 9). The next rows show the samples causing most interference for the classifier (Eq. 1)

Generative models such as VAEs [16] are known to generate blurry images and images with mix of categories. To avoid such a source of noise in the optimization, we minimize an entropy penalty to encourage generating points for which the previous model is confident. The final objective of the generator based retrieval is

$$\max_{\boldsymbol{Z}} \sum_{z \in \boldsymbol{Z}} [D_{KL}(y_{pre} \parallel \hat{y}) - \alpha H(y_{pre})] \quad s.t. \quad ||z_i - z_j||_2^2 > \epsilon \, \forall z_i, z_j \in \boldsymbol{Z} \text{ with } z_i \neq z_j, \tag{2}$$

with the entropy $H$ and a hyperparameter $\alpha$ to weight the contribution of each term.

So far we have assumed having a perfect encoder/decoder that we use to retrieve the interfered samples from the previous history for the function being learned. Since we assume an online continual learning setting, we need to address learning the encoder/decoder continually as well.

We could use a variational autoencoder (VAE) with $p_\gamma(X \mid z) = \mathcal{N}(X \mid g_\gamma(z), \sigma^2 I)$ with mean $g_\gamma(z)$ and covariance $\sigma^2 I$.

As for the classifier we can also update the VAE based on incoming samples and the replayed samples. In Eq. 1 we only retrieve samples that are going to be interfered given the classifier update, assuming

a good feature representation. We can also use the same strategy to mitigate catastrophic forgetting in the generator by retrieving the most interfered samples given an estimated update of both parameters $(\phi, \gamma)$. In this case, the intereference is with respect to the VAE's loss, the evidence lower bound (ELBO). Let us denote $\gamma^v, \phi^v$ the virtual updates for the encoder and decoder given the incoming batch. We consider the following criterion for retrieving samples for the generator:

$$\max_{\mathbf{Z}_{\text{gen}}} \; \underset{z \sim q_{\phi^v}}{E} \left[ -log(p_{\gamma^v}(g_{\gamma^v}(\mathbf{Z}_{\text{gen}})|z)) \right] - \underset{z \sim q_{\phi'}}{E} \left[ -log(p_{\gamma'}(g_{\gamma'}(\mathbf{Z}_{\text{gen}})|z)) \right]$$
$$+ D_{KL}(q_{\phi^v}(z|g_{\gamma^v}(\mathbf{Z}_{\text{gen}}))||p(z)) - D_{KL}(q_{\phi'}(z|g_{\gamma'}(\mathbf{Z}_{\text{gen}}))||p(z)) \qquad (3)$$
$$s.t. \quad ||z_i - z_j||_2^2 > \epsilon \; \forall z_i, z_j \in \mathbf{Z}_{\text{gen}} \; \text{s.t.} \; z_i \neq z_j$$

Here $(\phi', \gamma')$ can be the current VAE or stored from the end of the previous task. Similar to $\mathbf{Z}$, $\mathbf{Z}_{\text{gen}}$ is initialized with $Z_{\text{gen}} \sim q_\phi(X_t)$ and limited to few gradient updates. A complete view of the MIR based generative replay is shown in Algorithm 2

### 3.3 A Hybrid Approach

Training generative models in the continual learning setting on more challenging datasets like CIFAR-10 remains an open research problem [23]. Storing samples for replay is also problematic as it is constrained by storage costs and very-large memories can become difficult to search. To leverage the benefits of both worlds while avoiding training the complication of noisy generation, Similar to Riemer et al. [31] we use a hybrid approach where an autoencoder is first trained offline to store and compress incoming memories. Differently, in our approach, we perform MIR search in the latent space of the autoencoder using Eq. 1. We then select nearest neighbors from stored compressed memories to ensure realistic samples. Our strategy has several benefits: by storing lightweight representations, the buffer can store more data for the same fixed amount of memory. Moreover, the feature space in which encoded samples lie is fully differentiable. This enables the use of gradient methods to search for most interfered samples. We note that this is not the case for the discrete autoencoder proposed in [31]. Finally, the autoencoder with its simpler objective is easier to train in the online setting than a variational autoencoder. The method is summarized in Algorithm 3 in the Appendix.

---

**Algorithm 1:** Experience MIR (ER-MIR)

**Input:** Learning rate $\alpha$, Subset size $C$; Budget $\mathcal{B}$
1 **Initialize:** Memory $\mathcal{M}$; $\theta$
2 **for** $t \in 1..T$ **do**
3      **for** $B_n \sim D_t$ **do**
4          `%%Virtual Update`
5          $\theta^v \leftarrow \texttt{SGD}(B_n, \alpha)$
6          `%Select C samples`
7          $B_\mathcal{C} \sim \mathcal{M}$
8          `%Select based on score`
9          $S \leftarrow sort(s_{MI}(B_\mathcal{C}))$
10          $B_{\mathcal{M}_\mathcal{C}} \leftarrow \{S_i\}_{i=1}^\mathcal{B}$
11          $\theta \leftarrow SGD(B_n \cup B_{\mathcal{M}_\mathcal{C}}, \alpha)$
12          `%Add samples to memory`
13          $\mathcal{M} \leftarrow UpdateMemory(B_n);$
14      **end**
15 **end**

---

**Algorithm 2:** Generative-MIR (GEN-MIR)

**Input:** Learning rate $\alpha$
1 **Initialize:** Memory $\mathcal{M}$; $\theta, \phi, \gamma$
2 **for** $t \in 1..T$ **do**
3      $\theta', \phi', \gamma' \leftarrow \theta, \phi, \gamma$
4      **for** $B_n \sim D_t$ **do**
5          `%Virtual Update`
6          $\theta^v \leftarrow \texttt{SGD}(B_n, \alpha)$
7          $B_C \leftarrow$ Retrieve samples as per Eq (2)
8          $B_G \leftarrow$ Retrieve samples as per Eq (3)
9          `%Update Classifier`
10          $\theta \leftarrow SGD(B_n \cup B_C, \alpha)$
11          `%Update Generative Model`
12          $\phi, \gamma \leftarrow SGD(B_n \cup B_G, \alpha)$
13      **end**
14 **end**

---

## 4 Experiments

We now evaluate the proposed method under the generative and experience replay settings. We will use three standard datasets and the shared classifier setting described below.

- **MNIST Split** splits MNIST data to create 5 different tasks with non-overlapping classes. We consider the setting with 1000 samples per task as in [2, 26].
- **Permuted MNIST** permutes MNIST to create 10 different tasks. We consider the setting with 1000 samples per task as in [2, 26].

- **CIFAR-10 Split** splits CIFAR-10 dataset into 5 disjoint tasks as in Aljundi et al. [3]. However, we use a more challenging setting, with all 9,750 samples per task and 250 retained for validation.
- **MiniImagenet Split** splits MiniImagenet [38] dataset into 20 disjoint tasks as in Chaudhry et al. [8] with 5 classes each.

In our evaluations we will focus the comparisons of MIR to random sampling in the experience replay (ER) [3, 8] and generative replay [36, 22] approaches which our method directly modifies. We also consider the following reference baselines:

- **fine-tuning** trains continuously upon arrival of new tasks without any forgetting avoidance strategy.
- **iid online** (upper-bound) considers training the model with a single-pass through the data on the same set of samples, but sampled iid.
- **iid offline** (upper-bound) evaluates the model using multiple passes through the data, sampled iid. We use 5 epochs in all the experiments for this baseline.
- **GEM** [26] is another method that relies on storing samples and has been shown to be a strong baseline in the online setting. It gives similar results to the recent A-GEM [6].

We do not consider prior-based baselines such as Kirkpatrick et al. [18] as they have been shown to work poorly in the online setting as compared to GEM and ER [8, 26]. For evaluation we primarily use the accuracy as well as forgetting [8].

**Shared Classifier** A common setting for continual learning applies a separate classifier for each task. This does not cover some of the potentially more interesting continual learning scenarios where task metadata is not available at inference time and the model must decide which classes correspond to the input from all possible outputs. As in Aljundi et al. [3] we adopt a shared-classifier setup for our experiments where the model can potentially predict all classes from all tasks. This sort of setup is more challenging, yet can apply to many realistic scenarios.

**Multiple Updates for Incoming Samples** In the one-pass through the data continual learning setup, previous work has been largely restricted to performing only a single gradient update on incoming samples. However, as in [3] we argue this is not a necessary constraint as the prescribed scenario should permit maximally using the current sample. In particular for replay methods, performing additional gradient updates with additional replay samples can improve performance. In the sequel we will refer to this as performing more iterations.

**Comparisons to Reported Results** Note comparing reported results in Continual Learning requires great diligence because of the plethora of experimental settings. We remind the reviewer that our setting, i.e. shared-classifier, online and (in some cases) lower amount of training data, is more challenging than many of the other reported continual learning settings.

### 4.1 Experience Replay

Here we evaluate experience replay with MIR comparing it to vanilla experience replay [8, 3] on a number of shared classifier settings. In all cases we use a single update for each incoming batch, multiple iterations/updates are evaluated in a final ablation study. We restrict ourselves to the use of reservoir sampling for deciding which samples to store. We first evaluate using the MNIST Split and Permuted MNIST (Table 1). We use the same learning rate, 0.05, used in Aljundi et al. [3]. The number of samples from the replay buffer is always fixed to the same amount as the incoming samples, 10, as in [8]. For MIR we select by validation $C = 50$ and the $s_{MI\text{-}2}$ criterion for both MNIST datasets. ER-MIR performs well and improves over (standard) ER in both accuracy and forgetting. We also show the accuracy on seen tasks after each task sequence is completed in Figure 7.

We now consider the more complex setting of CIFAR-10 and use a larger number of samples than in prior work [3]. We study the performance for different memory sizes (Table 2). For MIR we select by validation at $M = 50$, $C = 50$ and the $s_{MI\text{-}1}$ criterion. We observe that the performance gap increases when more memories are used. We find that the GEM method does not perform well in this

|  | Accuracy ↑ | Forgetting ↓ |  | Accuracy ↑ | Forgetting ↓ |
|---|---|---|---|---|---|
| iid online | $86.8 \pm 1.1$ | N/A | iid online | $73.8 \pm 1.2$ | N/A |
| iid offline | $92.3 \pm 0.5$ | N/A | iid offline | $86.6 \pm 0.5$ | N/A |
| fine-tuning | $19.0 \pm 0.2$ | $97.8 \pm 0.2$ | fine-tuning | $64.6 \pm 1.7$ | $15.2 \pm 1.9$ |
| GEN | $79.3 \pm 0.6$ | $19.5 \pm 0.8$ | GEN | $79.7 \pm 0.1$ | $5.8 \pm 0.2$ |
| GEN-MIR | $\mathbf{82.1 \pm 0.3}$ | $\mathbf{17.0 \pm 0.4}$ | GEN-MIR | $\mathbf{80.4 \pm 0.2}$ | $\mathbf{4.8 \pm 0.2}$ |
| GEM [26] | $86.3 \pm 1.4$ | $11.2 \pm 1.2$ | GEM [26] | $78.8 \pm 0.4$ | $\mathbf{3.1 \pm 0.5}$ |
| ER | $82.1 \pm 1.5$ | $15.0 \pm 2.1$ | ER | $78.9 \pm 0.6$ | $3.8 \pm 0.6$ |
| ER-MIR | $\mathbf{87.6 \pm 0.7}$ | $\mathbf{7.0 \pm 0.9}$ | ER-MIR | $\mathbf{80.1 \pm 0.4}$ | $3.9 \pm 0.3$ |

Table 1: Results for MNIST SPLIT (left) and Permuted MNIST (right). We report the Average Accuracy (higher is better) and Average Forgetting (lower is better) after the final task. We split results into priveleged baselines, methods that don't use a memory storage, and those that store memories. For the ER methods, 50 memories per class are allowed. Each approach is run 20 times.

|  | Accuracy ↑ | | | Forgetting ↓ | | |
|---|---|---|---|---|---|---|
|  | $M = 20$ | $M = 50$ | $M = 100$ | $M = 20$ | $M = 50$ | $M = 100$ |
| iid online | $60.8 \pm 1.0$ | $60.8 \pm 1.0$ | $60.8 \pm 1.0$ | N/A | N/A | N/A |
| iid offline | $79.2 \pm 0.4$ | $79.2 \pm 0.4$ | $79.2 \pm 0.4$ | N/A | N/A | N/A |
| GEM [26] | $16.8 \pm 1.1$ | $17.1 \pm 1.0$ | $17.5 \pm 1.6$ | $73.5 \pm 1.7$ | $70.7 \pm 4.5$ | $71.7 \pm 1.3$ |
| iCarl (5 iter) [30] | $28.6 \pm 1.2$ | $33.7 \pm 1.6$ | $32.4 \pm 2.1$ | $\mathbf{49 \pm 2.4}$ | $40.6 \pm 1.1$ | $40 \pm 1.8$ |
| fine-tuning | $18.4 \pm 0.3$ | $18.4 \pm 0.3$ | $18.4 \pm 0.3$ | $85.4 \pm 0.7$ | $85.4 \pm 0.7$ | $85.4 \pm 0.7$ |
| ER | $27.5 \pm 1.2$ | $33.1 \pm 1.7$ | $41.3 \pm 1.9$ | $50.5 \pm 2.4$ | $35.4 \pm 2.0$ | $23.3 \pm 2.9$ |
| ER-MIR | $\mathbf{29.8 \pm 1.1}$ | $\mathbf{40.0 \pm 1.1}$ | $\mathbf{47.6 \pm 1.1}$ | $50.2 \pm 2.0$ | $\mathbf{30.2 \pm 2.3}$ | $\mathbf{17.4 \pm 2.1}$ |

Table 2: CIFAR-10 results. Memories per class $M$, we report (a) Accuracy, (b) Forgetting (lower is better). For larger sizes of memory ER-MIR has better accuracy and improved forgetting metric. Each approach is run 15 times.

setting. We also consider another baseline iCarl [30]. Here we boost the iCarl method permitting it to perform 5 iterations for each incoming sample to maximize its performance. Even in this setting it is only able to match the experience replay baseline and is outperformed by ER-MIR for larger buffers.

|  | Number of iterations | |
|---|---|---|
|  | 1 | 5 |
| iid online | $60.8 \pm 1.0$ | $62.0 \pm 0.9$ |
| ER | $41.3 \pm 1.9$ | $42.4 \pm 1.1$ |
| ER-MIR | $47.6 \pm 1.1$ | $49.3 \pm 0.1$ |

Table 3: CIFAR-10 accuracy (↑) results for increased iterations and 100 memories per class. Each approach is run 15 times.

|  | Accuracy ↑ | Forgetting ↓ |
|---|---|---|
| ER | $24.7 \pm 0.7$ | $23.5 \pm 1.0$ |
| ER-MIR | $25.2 \pm 0.6$ | $18.0 \pm 0.8$ |

Table 4: MinImagenet results. 100 memories per class and using 3 updates per incoming batch, accuracy is slightly better and forgetting is greatly improved. Each approach is run 15 times

**Increased iterations** We evaluate the use of additional iterations on incoming batches by comparing the 1 iteration results above to running 5 iterations. Results are shown in Table 3 We use ER an and at each iteration we either re-sample randomly or using the MIR criterion. We observe that increasing the number of updates for an incoming sample can improve results on both methods.

**Longer Tasks Sequence** we want to test how our strategy performs on longer sequences of tasks. For this we consider the 20 tasks sequence of MiniImagenet Split. Note that this dataset is very challenging in our setting given the shared classifier and the online training. A naive experience replay with 100 memories per class obtains only 17% accuracy at the end of the task sequence. To overcome this difficulty, we allow more iterations per incoming batch. Table 4 compares ER and ER-MIR accuracy and forgetting at the end of the sequence. It can be seen how our strategy continues to outperform, in particular we achieve over 5% decrease in forgetting.

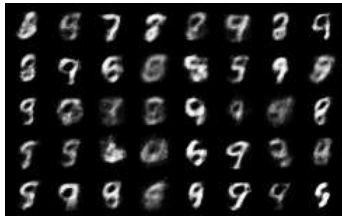

(a) Generation with the best VAE baseline. Complications arising from both properties leave the VAE generating blurry and/or fading digits.

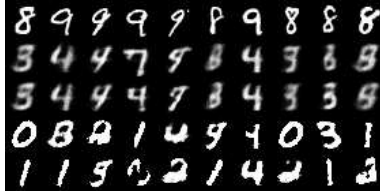

(b) Most interfered samples while learning the last task (8 vs 9). Top row is the incoming batch. Rows 2 and 3 show the most interfered samples for the classifier, Row 4 and 5 for the VAE. We observe retrieved samples look similar but belong to different category.

Figure 3: Online and low data regime MNIST Split generation. Qualitatively speaking, most interfered samples are superior to baseline's.

## 4.2 Generative Replay

We now study the effect of our proposed retrieval mechanism in the generative replay setting (Alg. 2). Recall that online continual generative modeling is particularly challenging and to the best of our knowledge has never been attempted. This is further exacerbated by the low data regime we consider.

Results for the MNIST datasets are presented in Table 1. To maximally use the incoming samples, we (hyper)-parameter searched the amount of additional iterations for both GEN and GEN-MIR. In that way, both methodologies are allowed their optimal performance. More hyperarameter details are provided in Appendix B.2. On MNIST Split, MIR outperforms the baseline by $2.8\%$ and $2.5\%$ on accuracy and forgetting respectively. Methods using stored memory show improved performance, but with greater storage overhead. We provide further insight into theses results with a generation comparison (Figure 3). Complications arising from online generative modeling combined with the low data regime cause blurry and/or fading digits (Figure 3a) in the VAE baseline (GEN). In line with the reported results, the most interfered retrievals seem qualitatively superior (see Figure 3b where the GEN-MIR generation retrievals is demonstrated). We note that the quality of the samples causing most interference on the VAE seems higher than those on the classifier.

For the Permuted MNIST dataset, GEN-MIR not only outperforms the its baselines, but it achieves the best performance over all models. This result is quite interesting, as generative replay methods can't store past data and require much more tuning.

The results discussed thus far concern classification. Nevertheless, GEN-MIR alleviates catastrophic forgetting in the generator as well. Table 5 shows results for the online continual generative modeling. The loss of the generator is significantly lower on both datasets when it rehearses on maximally interfered samples versus on random samples. This result suggest that our method is not only viable in supervised learning, but in generative modeling as well.

Our last generative replay experiment is an ablation study. The results are presented in Table 6. All facets of our proposed methodology seem to help in achieving the best possible results. It seems however that the minimization of the label entropy, i.e. $H(y_{pre})$, which ensures that the previous classifier is confident about the retrieved sample's class, is most important and is essential to outperform the baseline.

|  | MNIST Split | Permuted MNIST |
|---|---|---|
| GEN | $107.2 \pm 0.2$ | $196.7 \pm 0.7$ |
| GEN-MIR | $\mathbf{102.5 \pm 0.2}$ | $\mathbf{193.7 \pm 1.0}$ |

Table 5: Generator's loss ($\downarrow$), i.e. negative ELBO, on the MNIST datasets. Our methodology outperforms the baseline in online continual generative modeling as well.

As noted in [23], training generative models in the continual learning setting on more challenging datasets remains an open research problem. [23] found that generative replay is not yet a viable strategy for CIFAR-10 given the current state of the generative modeling. We too arrived at the same conclusion, which led us to design the hybrid approach presented next.

|                              | Accuracy |
|------------------------------|----------|
| GEN-MIR                      | 83.0     |
| ablate MIR on generator      | 82.7     |
| ablate MIR on classifier     | 81.7     |
| ablate $D_{KL}(y_{pre} \parallel \hat{y})$ | 80.7 |
| ablate $H(y_{pre})$          | 78.3     |
| ablate diversity constraint  | 80.7     |
| GEN                          | 80.0     |

Table 6: Ablation study of GEN-MIR on the MNIST Split dataset. The $H(y_{pre})$ term in the MIR loss function seems to play an important role in the success of our method.

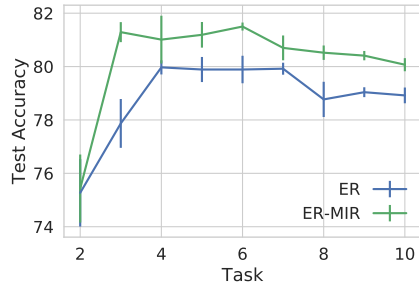

Table 7: Permuted MNIST test accuracy on tasks seen so far for rehearsal methods.

## 4.3 Hybrid Approach

In this section, we evaluate the hybrid approach proposed in Sec 3.3 on the CIFAR-10 dataset. We use an autoencoder to compress the data stream and simplify MIR search.

We first identify an important failure mode arising from the use of reconstructions which may also apply to generative replay. During training, the classifier sees real images, from the current task, from the data stream, along with reconstructions from the buffer, which belong to old tasks. In the shared classifier setting, this discrepancy can be leveraged by the classifier as a discriminative feature. The classifier will tend to classify all real samples as belonging to the classes of the last task, yielding low test accuracy. To address this problem, we first autoencode the incoming data with the generator before passing it to the

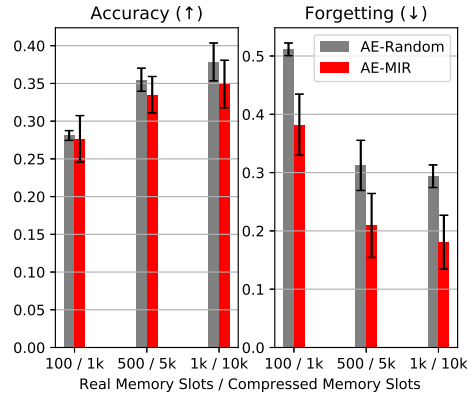

Figure 4: Results for the Hybrid Approach

classifier. This way, the classifier cannot leverage the distribution shift. We found that this simple correction led to a significant performance increase. We perform an ablation experiment to validate this claim, which can be found in Appendix C, along with further details about the training procedure.

In practice, we store a latent representation of size $4 \times 4 \times 20 = 320$, giving us a compression factor of $\frac{32 \times 32 \times 3}{320} = 9.6$ (putting aside the size of the autoencoder, which is less than 2% of total parameters for large buffer size). We therefore look at buffer size which are 10 times as big i.e. which can contain 1k, 5k, 10k compressed images, while holding memory equivalent to storing 100 / 5000 / 1k real images. Results are shown in Figure 4. We first note that as the number of compressed samples increases we continue to see performance improvement, suggesting the increased storage capacity gained from the autoencoder can be leveraged. We next observe that even though AE-MIR obtains almost the same average accuracies as AE-Random, it achieved a big decrease in the forgetting metric, indicating a better trade-offs in the performance of the learned tasks. Finally we note a gap still exists between the performance of reconstructions from incrementally learned AE or VAE models and real images, further work is needed to close it.

## 5 Conclusion

We have proposed and studied a criterion for retrieving relevant memories in an online continual learning setting. We have shown in a number of settings that retrieving interfered samples reduces forgetting and significantly improves on random sampling and standard baselines. Our results and analysis also shed light on the feasibility and challenges of using generative modeling in the online continual learning setting. We have also shown a first result in leveraging encoded memories for more compact memory and more efficient retrieval.

## Acknowledgements

We would like to thank Kyle Kastner and Puneet Dokania for helpful discussion. Eugene Belilvosky is funded by IVADO and Rahaf Aljundi is funded by FWO.

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
