[Supplementary Material]

|  | M=20 | M=50 | M=100 | M=20 | M=50 | M=100 |
|---|---|---|---|---|---|---|
| iid online | $86.8 \pm 1.1$ | $86.8 \pm 1.1$ | $86.8 \pm 1.1$ | N/A | N/A | N/A |
| iid offline | $86.8 \pm 1.1$ | $86.8 \pm 1.1$ | $86.8 \pm 1.1$ | N/A | N/A | N/A |
| single | $19 \pm 0.2$ | $19 \pm 0.2$ | $19 \pm 0.2$ | $97.8 \pm 0.2$ | $97.8 \pm 0.2$ | $97.8 \pm 0.2$ |
| ER | $78.9 \pm 1.5$ | $82.6 \pm 1.0$ | $81.3 \pm 1.9$ | $19.1 \pm 2.0$ | $14.0 \pm 1.1$ | $15.8 \pm 2.7$ |
| ER-MIR | $81.5 \pm 1.9$ | $\mathbf{87.4 \pm 0.8}$ | $\mathbf{87.4 \pm 1.2}$ | $15.9 \pm 2.5$ | $\mathbf{7.2 \pm 0.9}$ | $\mathbf{6.7 \pm 1.4}$ |

Table 4: MNIST results. Memories per class $M$, we report the (a) Accuracy (b) Forgetting (lower is better). For larger sizes of memory ER-MIR has better accuracy and improved forgetting metric. Each approach is run 20 times

|  | M=20 | M=50 | M=100 | M=20 | M=50 | M=100 |
|---|---|---|---|---|---|---|
| iid online | $73.8 \pm 1.2$ | $73.8 \pm 1.2$ | $73.8 \pm 1.2$ | N/A | N/A | N/A |
| iid offline | $86.6 \pm 0.5$ | $86.6 \pm 0.5$ | $86.6 \pm 0.5$ | N/A | N/A | N/A |
| single | $64.6 \pm 1.7$ | $64.6 \pm 1.7$ | $64.6 \pm 1.7$ | $15.2 \pm 1.9$ | $15.2 \pm 1.9$ | $15.2 \pm 1.9$ |
| ER | $76.3 \pm 0.6$ | $78.4 \pm 0.5$ | $79.9 \pm 0.3$ | $5.6 \pm 0.6$ | $3.7 \pm 0.5$ | $2.48 \pm 0.5$ |
| ER-MIR | $76.3 \pm 0.5$ | $\mathbf{80.1 \pm 0.3}$ | $\mathbf{82.3 \pm 0.2}$ | $6.5 \pm 0.5$ | $3.4 \pm 0.3$ | $\mathbf{1.89 \pm 0.3}$ |

Table 5: Permuted MNIST results. Memories per class $M$, we report the (a) Accuracy (b) Forgetting (lower is better). For larger sizes of memory ER-MIR has better accuracy and improved forgetting metric. Each approach is run 10 times

# A    Full Results on ER-MIR

In this section we show full results on the ER-MIR for different settings of the buffer size for Permuted MNIST and MNIST Split. We also include results for CIFAR-10 with 1000 samples per task as studied in [3]. We note that the margins of gain for ER-MIR is lower here than in the full CIFAR-10 setting (using 9750 samples per task) suggesting ER-MIR is more effective in the more challenging settings.

# B    Details of Hyperparameters

## B.1    Experience Replay Experiments

For ER and ER-MIR we use the same base settings as in [3, 8]. Specifically the batch size is 10 for the ncoming samples and 10 for the buffered samples. As in that work we use a learning rate of 0.05 for our MNIST experiments. For CIFAR-10 we select by validation 0.1. ER-reservoir-MIR we also add the hyperparameter of the initial sampling size, $C$, which is chosen from $30, 50, 100, 150$ to be 50.

For MNIST we use a 2 layer MLP with 400 hidden nodes. For CIFAR-10 experiments we use a standard Resnet-18 used in [25, 6].

## B.2    Generative Modeling Experiments

Regarding the VAE used for generative replays: the encoder/decoder are 2 layers gated MLP networks with 400 hidden nodes and ReLU activations; the latent space size is 50 dimensions for MNSIT Split and 100 for Permuted MNSIT. To achieve the best possible baseline (GEN), here are some hyperameters we searched for: learning rate, dropout, the weight of the $KL(q(z|x)||p(z))$ in the loss and KL cost annealing shcedules. For GEN-MIR, we also searched for the weights of each loss in our proposed solutions e.g. $\lambda$. To keep things fair, GEN and GEN-MIR where allowed the same amount of trials.

# C    Further Description of Hybrid Approach

We give the algorithm block fully describing the method of Sec. 3.3.

---

**Algorithm 3:** AE-MIR

---

**Input:** Learning rate $\alpha$, Subset size $C$; Budget $\mathcal{B}$, Gen. Epochs $N_{\text{gen}}$

**1 Initialize:** Memory $\mathcal{M}$; $\theta, \theta_{ae}$

**2 for** $t \in 1..T$ **do**

    **3**    %%Offline Generator Training

    **4**    **for** $epoch \in 1...N_{gen}$ **do**

    **5**        **for** $B_n \sim D_t$ **do**

    **6**          $h \;\;\leftarrow \text{Encode}(\theta_{ae}; B_n)$

    **7**          $\tilde{B}_n \leftarrow \text{Decode}(\theta_{ae}; h)$

    **8**          $\text{loss}_{ae} \leftarrow \text{MSE}(\tilde{B}_n, B_n)$

    **9**          $\text{Adam}(\text{loss}_{ae}, \theta_{ae})$

    **10**        **end**

    **11**   **end**

    **12**   **for** $B_n \sim D_t$ **do**

    **13**        %%Virtual Update

    **14**        $\theta_v \leftarrow \text{SGD}(B_n, \alpha)$

    **15**        %%Autoencode batch

    **16**        $h \;\;\leftarrow \text{Encode}(\theta_{ae}; B_n)$

    **17**        $\tilde{B}_n \leftarrow \text{Decode}(\theta_{ae}; h)$

    **18**        %Select C samples

    **19**        $B_{\mathcal{C}} \sim \mathcal{M}$

    **20**        $B_G \leftarrow$ Retrieve samples acc. to Eq 1

    **21**        %%Store compressed rep.

    **22**        $\mathcal{M} \leftarrow \text{UpdateMemory}(h, L_n)$

    **23**        %% Train the Classifier

    **24**        $\theta \leftarrow SGD(\tilde{B}_n \cup B_{\mathcal{M}_{\mathcal{C}}}, \alpha)$

    **25**   **end**

**26 end**

---

For all experiments, we train the generator offline for 5 epochs, but still in the incremental setting. As in the replay experiments, the batch size is 10. All results are averaged over 5 runs.

**Ablation Study**    Here we provide results for the AE hybrid approach. We first change the test set evaluation, by feeding the real images, instead of autoencoded ones. We denote this model as "- test AE". We also look at additionally feeding real images from the current data stream, instead of reconstructed ones (i.e. replacing line 24 $\theta \leftarrow SGD(\tilde{B}_n \cup B_{\mathcal{M}_{\mathcal{C}}}, \alpha)$ as $\theta \leftarrow SGD(B_n \cup B_{\mathcal{M}_{\mathcal{C}}}, \alpha)$). We call this model "- train & test AE".

From these results we see that *never* training the classifier on real images is essential to obtain good results, as "- train & test AE" performs badly. Moreover, we notice that also autoencoding the data at

Figure 7: Ablation results

Figure 8: Reference Performance

423 test time is also responsible for some performance gain. This is denoted by the small but noticeable
424 performance increase from "- test AE" to "AE-Random"