[Reviews · NeurIPS 2019]

Reviewer 1



The authors propose a new way to use replay to avoid catastrophic forgetting. What the authors propose is after each new example to train on the previous examples that had the loss go up the most after the update. This is a very simple and it is clear why this should be better than replay uniform at random. In order to make their algorithm feasible the authors first select uniformly random subset of C examples, then they have to compute how the loss changed between all those C example due to the parameter update and then they do a training step on the K examples where the loss changed most. As a way to make this more efficient the authors propose to instead do nearest neighbor search in the latent space of an autoencoder to find similar examples. This method reminds me of memory based parameter adaptation by sprechmann et al (https://arxiv.org/abs/1802.10542) where the authors also do continual learning by doing nearest neighbor search in a latent space. I am a bit worried about this approach in so far as when the new task is very dissimilar to the old task the examples retrieved by lookup in autoencoder latent space could well end up being quite meaningless, however the nearest neighbor search will give examples which are not very diverse. In general it seems to me using the nearest neighbor search in autoencoder latent space would easily give examples which are not very diverse. Have the authors observed this in practice? Could you comment on whether you see this as a problem? The experiments seem fairly toyish to me, however I think the idea is nice and should be published

Reviewer 2



This paper describes an approach to improve rehearsal-based continual learning techniques (either replay-based or with a generative model) by identifying samples that are most useful to avoid forgetting. This is achieved by computing the increase in loss on the replayed samples, and using this to determine which samples should be used during learning. It is a simple and intuitive idea, the paper is clearly written, and experiments on multiple datasets are compelling. I think it could make a nice addition to the conference, but needs a few improvements first. My main criticism is that the approach requires a separate virtual gradient step for each actual step, to compute the change in loss on the replay samples. I think the discussion around this could be stronger, eg. explicitly mentioning the overhead in computation / memory for this approach versus others. Is there a way to alleviate this additional step (eg. keep a running estimate of 'utility' for each sample based on previous gradient steps)? If not, a fairer comparison with baselines should include twice as many gradient steps (though still online). Does this make any difference? A few other comments and questions: - Why is the hybrid approach in the appendix rather than the main experiments? It seems like one of the central proposed methods to this paper - and given it's introduced in the main body, the experiments should appear as well. - The performance of baselines in Split MNIST seems poorer than that reported in related work (eg. iid offline is 97, and Deep Generative Replay (which is not included in this paper as a comparison) is at 92. Is this because of multiple vs a single pass? Additional baselines and clarification would be useful here. - The only external comparison is with GEM and, effectively, DGR (the GEN case), with the rationale that prior-based approaches like EWC don't perform as well, but I think other performant methods should be included (eg. VCL). This may not require reimplementing the approach, but comparing with the same conditions as previously published results. - In the single-pass setting, things like learning rate become more important - how were hyperparameter sweeps for baselines performed? - Some of the reproducibility checklist items are missing, and are easy to fix (eg. description of compute) POST-REBUTTAL: After reading the response and other reviews, I think the authors have done a great job of addressing my concerns. The additional baselines (both on the side of additional gradient steps and on other approaches like VCL) strengthen the paper. The discussion around gradient steps and computation provided by the authors in the rebuttal is convincing, and I think this should appear in the camera-ready version. i.e. comparison of computation / gradient steps taken by other approaches, and the 2-step baseline to show that this doesn't help with a standard approach. I think this is a good paper, and I have increased my score to a 7.

Reviewer 3



Originality: This paper considers generative modeling in the online continual learning setting for the first time. While previous algorithms in continual learning are based on random sampling to store the past samples, the authors proposed more sophisticated method to construct the memory structure. Moreover, the authors treats Hybrid approach of experience replay and generative replay to obtain the benefits of both. Quality : The authors do not provide theoretical guarantee for the proposed algorithms. Although the experimental results show its effectiveness compared to the existing state-of-the-art algorithms, the proposed methods are not applied to longer task sequence than 10 different tas and real-world data beyond relatively easy benchmark dataset. Clarity: Although this paper is well-organized, there are some concerns about the clarity. What is the definition of the sample loss l in Section 3.1? Do not it have two variables like l(f_\theta(x),y)? Similarly, do not \mathcal{L} have two variables in equation (1)? I do not understand the meaning of “Single trains the model …” in line 181. Significance: This paper has great significance because this paper considers generative modeling in the online continual learning setting for the first time as stated above. In this sense, this paper address a difficult task which was not treated in the previous studies. Moreover, the authors propose a better sampling method for memory structure.

[Author Response · NeurIPS 2019]

We thank the reviewers for their work, comments, relevant questions, and generally positive feedback, addressed in turn:

**[R1] Sprechmann et al [S]**, thank you for this reference we will add it. Key differences: [S] do retrieval at inference
time (vs training), our method is *not* retrieving nearest neighbors, but uses the more sophisticated MIR criteria. The
experiments in [S] are more limited as the retrieval is done in pixel space ([S] Sec. 4.1 paragraph 3)
**case of dissimilar tasks** We are not certain we understood this criticism correctly. For clarification, the only nearest
neighbor lookup is done in the hybrid method. We employ it to find the training datapoint that are the closest (in latent
space) to the ones retrieved via MIR optimization. If the MIR criteria retrieves diverse latent codes, then the nearest
neighbor lookup will find diverse samples as well. In Permuted MNIST, samples have very different appearance and we
were still able to improve over compared methods.
**diversity** We use a diversity penalty (L113-115) in Generative MIR. L246-250 and Figure 5 further studies the effects
of the diversity penalty. In ER-MIR, diversity is enforced via sampling prior to applying the criterion (L102-104).
**more challenging data** we note these datasets are also used in the related work on this challenging online continual
learning with shared classifier setting. We now extend our ER-MIR experiments to Mini-ImageNet split. We train on 17
tasks as in [6] but with shared-head. Over 20 runs we obtain an accuracy of *26.4%±0.6* vs ER accuracy of *25.5%± 0.7*
and a substantial gain in forgetting (*19.1%± 0.8* vs *23.5%± 1.2* for ER baseline).

**[R2] additional compute of a separate virtual gradient step. comparison with baselines with twice as many**
**gradient steps. overhead/memory of this approach** For Generative MIR the number of online updates is a hyper-
parameter we searched for both GEN-ER baseline and our GEN-MIR (see Appendix B.2), thus in both cases adding
more gradient steps would not help. Indeed in the online setting there is not a clear correlation between more iterations
on the incoming data (and buffer) and performance, as more iterations leads to more forgetting and overfitting. We
add an ablation on Split MNIST where we count the virtual update as an iteration i.e. the random baseline is allowed
2x more real updates –> 2 iterations: GEN *57.4%* / GEN-MIR *82.1%*, 10 iterations: GEN *76.8%* / GEN-MIR *83.3%*,
100 iterations GEN *70.7%* / GEN-MIR: *69.7%*. Note that the regular GEN is much more sensitive to using very few
iterations. For ER-MIR experiments only 1 update is done in our experiments except in Table 3, observe there that even
with 5 gradient steps for ER, ER-MIR with 1 is still superior. We emphasize our work's aim was to determine if the
non-uniform sampling strategy works versus uniform, which we have shown it generally is, a critical first step to future
work that can find more efficient approximation of the criteria. Notably existing continual learning methods can indeed
be computationally very expensive compared to standard training methods. For example GEM can be 10x slower than
ER and other methods[6]. Our un-optimized ER-MIR implementation (for CIFAR-10) is approximately 3x slower in
wall clock time than regular ER. In terms of memory consumption it is the same as ER with equivalent buffer.
**Why is hybrid approach in the appendix?** The primary results of the hybrid experiments are indeed in the main
paper (see Fig 6 and L281-285). Due to space constraints we put the (sizable) algorithm block of the hybrid method as
well as the ablation study in the appendix.
**"Performance of baselines seems poorer than related work"** Our experiments focus on the online setting (e.g.
[3,6,26])with a shared head classifier (see [3,9]). The accuracy for DGR you refer to is for offline and multi-headed
evaluation aka at test time the classification only chooses between 2 categories versus 10 and is thus a very different
setting. Note that our baselines are similar to those reported in [3] which also considers online and shared classifier.
Details of all setups are given in experiments and appendix, code is included in supplementary material and will be
further extended in release.
**More external comparisons (suggested VCL).** GEM and ER are the state of the arts in the online continual learning
case. For EWC [1] and [2] report extremely poor performance in this setting compared GEM/ER. We have added an
additional comparison to VCL. Using the official VCL code, and the same experimental setup as ours (online, single
head, 1k samples per task, 50 buffer slots per class) we ran VCL on Split MNIST and Permuted MNIST. VCL with
buffer gives (*87.2%, 65.9%*) and without (*73.9%, 64.4%*), on Split and Permuted MNIST respectively. These results fall
below the MIR performance (*87.6%, 80.1%*) by *0.4%* and *14.2%*. We also tried VCL with 2 gradients steps, however
this did not help performance. Note the buffer of VCL is used after training with an additional offline training steps
on it. Before any prediction step during learning, training on the buffer has to be performed which violates the online
continual learning setting we consider in this paper. Moreover, VCL is orthogonal to MIR, and both can be used jointly.
**Hyperparameter sweeps for baseline, checklist** We use a validation set as described in Sec 4 and Appendix for
baseline and our method. In terms of learning rates ranges are similar to those used in other works [5,7]. For compute
we utilized a single GPU in all experiments. We reiterate that all methods were given the same number of trial runs.

**[R3] Theory** Unfortunately there is very little existing tools for theoretical understanding of the continual learning
methods (and especially in the typical non-convex setting) much less the online counterparts to provide a basis for
analyzing MIR.
**Clarity** We will correct the ambiguous notation/text you note.
**Longer sequence** See response to R1 for new comparisons on Mini-ImageNet, we note the online and shared-head
setting is very challenging and longer sequences lead to extreme forgetting even with strong baselines like ER.

[Meta-Review · NeurIPS 2019]

Interesting approach to improve online continual learning (either based on replay or with a generative model) by identifying samples that are most useful to avoid forgetting.